# Relationships among Indicators of Metabolism, Mammary Health and the Microbiomes of Periparturient Holstein Cows

**DOI:** 10.3390/ani12010003

**Published:** 2021-12-21

**Authors:** Daniela C. Tardón, Christian Hoffmann, Fernanda C. R. Santos, Nathalia Decaris, Filipe A. Pinheiro, Luciano L. Queiroz, David J. Hurley, Viviani Gomes

**Affiliations:** 1Department of Internal Medicine, College of Veterinary Medicine and Animal Science, University of São Paulo, Sao Paulo 05508-270, Brazil; dcastrotardon.vet@gmail.com (D.C.T.); fe.carolinaramos@gmail.com (F.C.R.S.); nathalia.decaris@usp.br (N.D.); filipe.aguera@gmail.com (F.A.P.); 2Escuela de Medicina Veterinaria, Facultad de Recursos Naturales y Medicina Veterinaria, Universidad Santo Tomás, Santiago 3473620, Chile; 3Department of Food Sciences and Experimental Nutrition, School of Pharmaceutical Sciences, University of São Paulo, Sao Paulo 05508-000, Brazil; c.hoffmann@usp.br (C.H.); lqueiroz@usp.br (L.L.Q.); 4Microbiology Graduate Program, Department of Microbiology, Institute of Biomedical Science, University of São Paulo, Sao Paulo 05508-900, Brazil; 5Food Animal Health and Management Program, College of Veterinary Medicine, University of Georgia, Athens, GA 30602, USA; djhurley@uga.edu

**Keywords:** cattle, microbiota, transition period, metabolic biomarkers

## Abstract

**Simple Summary:**

Parturition is the most important physiological event in the lifecycle of dairy cows; it mediates changes in the microbiota composition. However, the complete picture of the dynamics of these phenomena and how they affect health and metabolism is unknown. This study documents the composition of the microbiota in the mammary gland, on reproductive surfaces and those associated with the rectum immediately after parturition. The microbiomes of different maternal niches were different, as predicted by their different functional roles in cows. Based on the results of this research, the conclusion that the microorganisms that colonize different mucosal tissues of cows were linked to the state of systemic energy metabolism and had an impact on the health of the mammary gland cows following calving was drawn.

**Abstract:**

During the period called “transition”, from the ceasing of milk production to the reestablishment of full milk production, it is postulated that the microbiota of cows undergo changes in composition driven by the fluxes in systemic energetics and that these changes appear to impact the health of cows. The primary objective of this study was to document the make-up of the microbiota in the mammary gland compared with those in the vagina and in feces in an attempt to determine any correlations between the composition of the microbiota, the impact of blood indicators of energetic metabolites and the health of the mammary gland at the time of calving. Samples were collected from 20 Holstein dairy cows immediately following calving to assess their general health and measure the microbiomes associated with each cow using 16S rRNA sequencing. The results indicated that the microbiomes found within each maternal niche were different. A set of significant negative associations between the blood energetic biomarkers (NEFAs, BHB, triglycerides and cholesterol) and the taxa *Pseudomonas*, *Christensenellaceae* and *Methanobrevibacter* were observed in this study. In contrast, *Escherichia* and *Romboutsia* were positively correlated with the same energetic metabolites. Therefore, it was concluded that there appears to be a set of relationships between the microorganisms that colonize several niches of cows and the sufficiency of systemic energy metabolism. Furthermore, both the microbiome and energy dynamics impact the health of the mammary gland of the host.

## 1. Introduction

The transition period in dairy cows is defined as the period from 3 weeks before calving to 3 weeks following calving. It is characterized by a series of adaptations in the metabolic and endocrine activities of cows. These support the energy requirements of pregnancy, calving and the onset of lactation. Part of the complex is mediated by a 30% reduction in dry matter consumption. The metabolic consequences of the change in energy demand and reduced energy consumption trigger a set of changes to blood metabolites that result from the mobilization of body reserves. Briefly, it has been reported that the elevation of serum non-esterified fatty acids (NEFAs) and β-hydroxybutyrate (BHB) was associated with decreased circulating glucose, cholesterol and triglycerides. The inability of cows to manage physiological adaptions results in the suppression of innate and adaptive immune activity as well as an enhanced risk of infectious diseases, episodes of mastitis and metritis in particular [1,2,3,4].

Because of the metabolic requirements of the mammary gland, particularly in early lactation, it has been suggested that a cow should be considered an appendage to the mammary gland rather than vice versa [5]. In the last third of pregnancy there are alterations in the secretory capacity of mammary epithelial cells. A decrease in parenchyma occurs gradually during lactation. The observed decrease may be induced by the metabolic conflict between pregnancy and lactation. A non-lactating period is necessary for the renewing of the gland to a state where optimal milk production in the succeeding lactation may occur; it is important for the replacement of senescent and damaged cells of the mammary epithelium [6]. Replacement cells may be responsible for the expansion and maintenance of the secretory cells of the mammary gland that influence colostrogenesis and milk production in the next lactation [6,7].

In ruminants, nutritional changes due to reduced feed intake during the periparturient period are associated with the modulation of the composition and function of the gut microbiota. This appears to result in a shift to the production of novel microbial-released metabolites by microorganisms on the surface of the gastrointestinal tract (GIT) [3,8]. Alterations in the GIT microbiome will, in turn, affect the systemic metabolic profile of cows [9]. Thus, the changes in the ecology or dynamics of the GIT and the associated changes in the circulating metabolite products are likely to impact both the GIT and systemic physiological functions; this is likely to alter the state of homeostasis. Altered nutrient availability and reduced available energy may also modulate the microbiota at other body sites. This would lead to both local and systemic consequences in functional homeostasis [9].

The microbiome and its metabolites have a crucial role in the maintenance of host homeostasis. Thus, the microbiome has become a critical area of research in both humans and animals [10]. The composition of the microbiota is of critical importance to the pool of nutrients available to the host. It fuels the processes that manage animal health as well as those that regulate tissue development in mammals, including cattle [11]. Multiple tissue-specific microbial communities exist within adult animals. The differences associated with specific mucosal tissue locations are governed by the specific physiological ecology of each niche. These are mediated by the nutrients available, the local pH, the transit rate of critical nutrients, the level of physiological activity within the space, the trafficking of microbes within the tissue and the immune cell populations present. Furthermore, the interactions that occur between host epithelial cells and colonizing symbiotic bacteria within the niche involved play a significant role in the local tissue environment, function and homeostasis [12]. The effects of local microbes on the management of host immune suppression and long-term tissue-specific colonization by specific bacterial species within a niche often result in disease when the balance in the tissue space is upset. This demonstrates the importance of understanding the interaction between the local tissue environments of the host and the population of colonizing microbes [13].

Currently, the profiles of microbes in different niches of cattle have been documented in previous studies [13,14,15,16,17,18,19]. However, there has been little emphasis on the make-up of the maternal microbiota at the time of calving or its relationship to the health of cows during the transition period. The aim of this study was to investigate any correlation between the composition of these microbiomes, the level of indicator blood metabolites and some health parameters in apparently healthy cows immediately after calving.

## 2. Materials and Methods

### 2.1. Animals and Management

The Committee on Ethics in the Use of Animals of FMVZ-USP (CEUA N° 2329260218) approved the experimental design and animal use for this study. The trial was carried out during the winter and spring of 2018 on a commercial farm located in Descalvado, São Paulo, Brazil (latitude: 21°54′14″ S; longitude: 47°37′12″ W).

Twenty multiparous Holstein cows, each in the 2nd to 5th lactation, were included in this study based on the date of calving, type of parturition (eutocic), volume of mammary secretion in the 1st milking (≥3 L) and colostrum quality (IgG ≥ 45 g/L).

All cows were kept on pasture during the period from day 60 to day 30 prior to calving. During the “closeup period” the cows were housed in a maternity compost barn system for a period of about 30 days prior to expected calving. The cows received the standard diet provided by the farm (defined in Appendix A). The diet met, or exceeded, the nutrimental and energy requirements for pre-calving dairy cows specified in the NRC guidelines (2001) [20]. The composition of the diet was prepared as a total mixed ration (TMR). This was offered twice daily. The main components of the ration were forage and corn silage. Supplemental minerals and vitamins were added during the feed-mixing process on the farm. Each cow consumed approximately 13 to 14 kg of dry matter (DM) per day. Access to water and mineral salt blocks were provided ad libitum during the whole pre-partum period.

All calvings were continuously monitored by members of the farm team or members of the research team for 24 h a day. The cows were transferred to maternity pens, a separate section within the barn, at the first signs of entering labor. These signs included behavioral changes, self-isolation, mucous discharge, vulvar edema and the relaxing of sacral ligaments [21]. After calving the delivery pens were restored to a state that was clean, dry, with good drainage and covered with fresh wood shavings and bedding material before the next calving. The samples for the assessments were collected as shown on the timeline presented in Figure 1.

### 2.2. Retrospective Data

Data from the farm record for each dam were collected from dairy record software (DelPro, DeLaval International AB, Tumba, Sweden). The values collected included the duration of the dry period prior to calving, the number of prior lactations and the total quantity of milk produced during the last lactation.

### 2.3. Collection of Colostrum

Immediately after calving (no longer than 0.5 h postpartum) the teats were cleaned and disinfected by employers of the farm using the standard operating procedure (SOP) of the farm. Mud and feces were removed from the udder using a soft brush. Next, each teat was immersed in a 2% chlorine bleach solution and dried with individual clean paper towels. A physical examination of the mammary gland was carried out by visual inspection with the palpation of teat structures [21]. A scoring system was utilized to provide numerical values for each finding, which is detailed in Appendix A. Next, the operator stripped the first jets of mammary secretion into a cup. The gross characteristics of the colostrum were assessed visually.

### 2.4. Collection and Preparation of Samples for Microbiome Assessment

Samples for analyses of the microbiomes associated with each site were harvested 0.5 h postpartum. Teats were again disinfected before a colostrum sample was collected. Each teat was sprayed with povidone iodine solution (Riodeine^®^, Rioquimica, São José do Rio Preto, SP, Brazil) and then dried with a sterile gauze. Furthermore, the surfaces of teat ends were rubbed with a sterile gauze saturated with 70% ethanol. For the microbiome assessment a volume of 50 mL of colostrum was collected into a sterile tube for each quarter (Falcon^®^, BD Biosciences, San Jose, CA, USA) by hand-milking while wearing a sterile glove. Once these samples were collected the whole gland was completely milked out. The volume of colostrum collected from each mammary gland was measured and the quality was determined using a colostrometer.

After the colostrum samples were collected the vaginal and fecal samples were collected. The external recto-vulvar region was cleaned and disinfected by being wiped with a dry paper towel to remove particulate contamination. Antiseptic cleaning of the surface with povidone iodine and washing with 70% ethanol followed the initial cleaning. The area was dried with a sterile gauze. A sample of vaginal secretion was obtained from an area close to the cervix using a long sterile swab in a movable a plastic sheath (Provar^®^, São Paulo, SP, Brazil). The tip of each swab was placed into a sterile DNAse-free microtube. This was repeated for each cow.

Fecal samples were obtained manually from the rectum using a sterile glove. The sample collected was desired to be about 20–50 g in size. Each sample was immediately deposited in a sterile plastic container. The samples to be utilized for microbiome analysis were held on dry ice (−70 °C) from the time of collection, during transportation from the farm and while held in the laboratory. Fecal samples were thawed and divided into smaller portions that fit into a 1.5 mL DNAse-free sterile tubes. Fecal sample manipulation was done under a sterile air laminar air flow hood.

The colostrum samples from all quarters of each cow were pooled after collection. A sample of five mL from each quarter was added to the other three in a new sterile 50 mL tube in the laboratory to yield a volume of 20 mL in the pool for each cow. These pools were each diluted 1:1 with 0.9% sterile saline. Two tubes containing diluted samples were centrifuged at 1500× *g* for 20 min and 20 °C. The supernatant was discarded and the centrifugation was repeated a second time. The cell pellet was collected from the bottom of each tube. The cells from both tubes were suspended in a total of 5 mL of sterile phosphate-buffered saline (PBS). The cells were stored in 1 mL portions in sterile DNAse-free micro-tubes. All fecal samples, vaginal swabs, diluted colostrum whey and residual colostrum cell samples were stored at −80 °C.

### 2.5. Colostrum Analysis

The remaining colostrum was used for the determination of the somatic cell count (SCC), a differential exam of the somatic cells, and to measure the IgG concentration of the colostrum. The SCC for each sample was determined using a direct microscopic method [22]. Briefly, colostrum samples were diluted 1:1 in phosphate-buffered saline (PBS) and 10 µL of the sample was spread over an area of one cm^2^. The samples were dried at room temperature for 24 h. The slides were fixed in methanol for 15 min and stained with the Rosenfeld dye [23]. The number of somatic cells was counted in 100 fields at 1000-times magnification using a standard brightfield optical microscope with an oil immersion (100×) objective. The total number of cells counted was multiplied by the area factor for the microscope field, the initial volume added to the slide (in this case 17,000) and the dilution of the sample (2) to obtain an estimate of the number of somatic cells per mL in the colostrum as collected. The SCC was determined for each cow in the pooled samples.

The differential of the cells found among the total somatic cells was performed using a cytocentrifugation method [22]. One mL of colostrum was diluted with 49 mL of PBS. This was centrifuged at 1200× *g* for 5 min at 4 °C. After centrifugation the colostrum separated into three distinct phases: a cell pellet, a portion of fluid whey and a layer of fat. The whey and fat were removed and not utilized. The cell pellet from each tube was diluted in 50 mL of PBS. The cells were washed three times. Finally, the cell pellet was suspended in 1 mL of RPMI 1640 (GIBCO, Sigma-Aldrich^®^, St. Louis, MO, USA). A sample of 100 µL of the cell suspension was centrifuged onto slides using a cytocentrifuge (CYTOSPIN 4, Thermo Scientific, Waltham, MA, USA).

These slides were fixed and stained by using the Rosenfeld method. The cellular morphology was analyzed by using a conventional brightfield microscope at 1000-times magnification. The leukocytes were classified as lymphocytes, monocytes, neutrophils, eosinophils or basophils based on cellular morphology and their staining pattern, as previously described (a minimum of 100 cells were counted per sample) [22]. The results acquired were expressed as the present amount of each cell type among those counted.

The concentration of IgG in serum from the cows was measured using a modification of a published sandwich ELISA [24]. The colostrum samples were centrifuged at 500× *g* for 10 min at 4 °C, yielding clean whey from each sample. The samples were used in the same ELISA as the serum. The samples were screened for both the order of magnitude of IgG in colostrum and to determine an endpoint value based on the measured order of magnitude as the starting point for the titration. Rabbit anti-bovine IgG antibody (B5645; Sigma, St. Louis, MO, USA) diluted 1:400 in a sodium carbonate buffer at pH 9.7 was used to coat Immulon 4HBX plates (Thermo Corp., Milford, MA, USA) at 4–8 °C overnight. The colostrum samples were diluted at 1:1,000,000 and 1:10,000,000 in an ELISA buffer (phosphate-buffered saline containing 0.5% Tween 20) for screening. The highest dilutions that indicated clearly IgG-positive samples for the screening assay were further diluted in a two-fold series to determine an endpoint titer (based on comparison with a positive control in each assay run). The samples were placed in duplicate wells for all assessments. The samples were incubated for 1 h at room temperature. The plates were washed three times with the ELISA buffer. Next, an IgG detection antibody—horseradish-peroxidase-conjugated rabbit anti-bovine IgG (A5295; Sigma, St. Louis, MO, USA)—was added to each well. The detection antibody was diluted 1:1000 in the ELISA buffer. All wells were incubated with the detection antibody for 30 min. The plates were washed three times with the ELISA buffer. The bound detection antibody was developed using 2,20-azino-bis (3-ethylbenzthiazoline-6-sulfonic acid) substrate (ABTS, A-9941; Sigma, St. Louis, MO, USA) containing a saturating concentration of hydrogen peroxide. The plates were incubated for 30 min to allow color development and measured using a plate reader with a 405 nm filter. The quantity of antibody in serum was determined relative to a six-point se-rial dilution of bovine gamma globulin standard over a range of 100 to 0.35 ng/mL (I5506; Sigma, St. Louis, MO, USA). The quantity of antibody in colostrum was similarly estimated.

### 2.6. Body Condition Score (BCS) Examination

At birth the weight of a cow was estimated using the thoracic diameter (heart girth) [25]. A team member then examined the cow and determined if there were any body anomalies. The BCS was evaluated immediately after calving on a five-point scale in 0.25-point increments [21].

### 2.7. Collection of Blood for Physiological Assessment and Serum Parameters

Blood samples were collected from each cow using the tail (coccygeal) vein. Samples were collected in vacutainer tubes containing sodium fluoride or without anticoagulant. These were used for the collection of plasma and serum, respectively. These samples were utilized to measure the metabolites needed for the assessment (non-esterified fatty acids, β-hydroxybutyrate, glucose, albumin, total protein, total serum iron, triglycerides, cholesterol and haptoglobin). Tubes of blood with no anticoagulant were clotted at 37 °C for 1 h and centrifuged at 1200× *g* for 10 min to obtain serum. The tubes of sodium fluoride blood were centrifuged at 1200× *g* for 10 min to obtain plasma. These samples were transferred to microtubes and stored in a freezer at −20 °C until they were assessed by the clinical laboratory.

### 2.8. Biochemical Marker Assessment

Plasma was used for the measurement of non-esterified fatty acids (NEFAs), β-hydroxybutyrate (BHB) and glucose. Albumin, total protein, total serum iron, triglyceride and cholesterol levels were determined from serum samples. Before assessment, the plasma or serum samples were thawed overnight at 4 °C. To provide homogenous samples the plasma or serum were vortexed for one minute prior to loading into the analytical instruments. The biochemical tests were performed using an automatic biochemical analyzer (Rx Daytona, Randox, Kearneysville, WV, USA) with commercial reagents according to the manufacturer’s instructions.

The concentration of haptoglobin (Hp) was determined by measuring relative Hp and meta-hemoglobin concentrations following published methods [26]. The standard curve was prepared using a serial dilution of control serum with established quantities of both Hp and meta-hemoglobin. The determination of serum haptoglobin concentration was calculated by the interpolation of a linear regression of the standard curve for each day’s assay set based on the absorbance at 450 nm on a microplate reader.

### 2.9. DNA Extraction, 16S rRNA Gene Amplification and High-Throughput Sequencing

Total DNA was extracted from all samples by utilizing a PowerSoil 96-well DNA Isolation Kit (MoBio Laboratories Inc., Carlsbad, CA, USA), strictly adhering to the manufacturer’s instructions. The V3–V4 region of the 16S rRNA gene was amplified using traditional PCR. The sequencing library preparation was carried out in a two-step PCR protocol, following Illumina’s recommendation. In the first PCR reaction we used the V3–V4 primers 341F–806R [27] as this pair has great taxonomy coverage in bacteria and archaea [28].

The PCR reactions were always carried out in triplicate using Platinum Taq (Invitrogen, Waltham, MA, USA) under the following conditions: 95 °C for 5 min, 25 cycles of 95 °C for 45 s, 55 °C for 30 s and 72 °C for 45 s and a final extension of 72 °C for 2 min for PCR 1. In PCR 2 the conditions were 95 °C for 5 min, 10 cycles of 95 °C for 45 s, 66 °C for 30 s and 72 °C for 45 s and a final extension of 72 °C for 2 min. The final PCR reaction was cleaned up using AMPureXP beads (Beckman Coulter, Brea, CA, USA) and samples were pooled in the sequencing libraries for quantification. The pool amplicon estimations were performed with Picogreen dsDNA assays (Invitrogen, Waltham, MA, USA) and the pooled libraries were then diluted for optimized qPCR quantification using a KAPA Library Quantification Kit for Illumina platforms (KAPA Biosystems, Woburn, MA, USA). The libraries were sequenced in a MiSeq 300 cycle (Illumina details) run system using the standard Illumina primers provided in the kit.

### 2.10. Bioinformatic and Statistical Analyses

The 16S rRNA sequences obtained from the microbiome were analyzed using the Quantitative Insights into Microbial Ecology (QIIME) v1.9.1 pipeline with default settings [27]. The reads were quality-filtered using Cutadapt, the forward sequences were cut at position 223 and the reverse sequences at position 98. These positions were chosen because the first quartile of the sequence quality score was greater than or equal to 20. Chimeric sequences were removed using UCHIME69 and operational taxonomic units (OTUs) using UCLUST70 with 97% similarity for bacterial sequences were considered for further analysis if they had a minimum of 5 sequences detected across all samples. Taxonomy was assigned to OTU representative sequences using the SILVA 16S database (version n132) [29]. All subsequent analyses were performed with R version 3.6.1 and RStudio version 3.6.1 statistical programs using phyloseq, vegan, gplots, ggplot2 and qiimer [30,31,32].

Samples were rarified to the lowest number (min. 1000) of sequences based on the alpha diversity analysis and the number of observed OTUs. The Shannon diversity index and Chao1 richness estimator were also calculated among samples. Beta diversity was calculated using weighted and unweighted UniFrac [33]. A heatmap was built using Ward’s hierarchical clustering method (ward.d2) and the OTUs were filtered with a minimum threshold of 1000.

The statistical evaluation of health parameters as correlated with the microbiome data were assessed as quantitative measures. The values were subjected to a test of their distribution relative to a Gaussian curve using the Shapiro–Wilk and Kolmogorov–Smirnov tests. Data that did not represent a normal distribution were analyzed using non-parametric methods. Spearman correlations were calculated to assess the correlation between OTUs from the colostrum, fecal and vaginal samples in addition to their association with the health parameters. This was only done for OTUs present in at least 10% of the samples. This was to reduce the number of multiple comparisons required and the associated compounding of error. *p* values were corrected using the false discovery rate method as implemented in the standard *p*.adjust function library of the master R base library. Only correlations with corrected *p* < 0.05 (*) were selected for the preparation of heat maps and included in the network analysis.

## 3. Results

### 3.1. Physiological Parameters

At calving, the dams in this study had a median summed udder clinical score of 9.00 ± 2.00 (Appendix A). The median value for the volume of colostrum was 6.00 ± 2.60 L. The median IgG concentration was 60.00 ± 10.45 g/L, as estimated by the colostrometer. The median IgG concentration measured by ELISA was 81.28 ± 19.62 mg/mL. The median SCC in colostrum was 2.20 ± 3.85 million cells/mL. The percentage of macrophages/epithelial cells, lymphocytes, neutrophils and eosinophils found in the colostrum secretion averaged 76.00%, 12.91%, 10.85% and 0.06%, respectively (Appendix A).

The BCS at calving time varied between 2.50 and 3.75 (median = 3.25) among the cows and their weight ranged from 607.00 to 856.00 kg (median = 778.00 kg). The values for the concentrations of NEFAs varied from 0.19 to 1.17 mmol/L (median = 0.64 mmol/L) and for BHB A from 0.01 to 0.47 mmol/L (median = 0.03 mmol/L). The glucose concentration ranged from 52 to 145 mg/d, with a median of 77.50 mg/dL. Cholesterol and triglyceride concentrations ranged between 38.10 and 105.90 mg/L (median = 70.20 mg/L) in addition to 3.6 and 81.5 mg/L (median = 29.55 mg/L), respectively. The serum values ranged from 2.87 to 8.4 g/dL (median = 6.40 g/dL) for total protein and 1.66 to 3.03 g/dL (median = 2.74) for albumin. The values for iron varied from 5.5 µmol/L to 37.90 µmol/L, with a median of 23.15 µmol/L. Finally, the haptoglobin values varied from 13 mg/dL to 59.03 mg/d, with a median of 22.92 mg/dL (Appendix A).

### 3.2. Descriptive Analysis of Microbiome

The quantitative estimates of bacterial phyla and genera in the colostrum, fecal and vaginal samples are shown in Figure 2. The most abundant phylum in the fecal and vaginal samples was Firmicutes, while the most frequently identified phylum in the colostrum samples was Proteobacteria.

Overall, at the genus taxonomic level the largest number of bacteria belonged to the *Christensenellaceae R-7* group, followed by lesser numbers of isolates from the genera *Paeniclostridium*, *Romboutsia*, *Tyzzerella* and *Ruminococcus* in the fecal and vaginal samples. The phylum Euryarchaeota, a genus *Methanobrevibacter* belonging to the Archaea kingdom, was abundant only in the fecal and vaginal samples. The colostrum samples presented a differential microbiological profile than either the vagina or fecal samples. They had a lower number of identified bacterial taxa than the other samples. The most frequent genera identified in these samples were *Pseudomonas*, *Staphylococcus* and *Acinetobacter*. Some colostrum samples had a higher frequency of *Ochrobactrum*.

In addition, the results in this figure allow us to visualize the fact that there is a similarity in OTUs and abundance between the fecal microbiomes and a group of vaginal samples. On the other hand, it is shown that another group of vaginal samples is similar to the microbial profile of colostrum.

The alpha diversity analysis indicated that the fecal samples showed greater richness and diversity in the bacterial community. The observed diversity was less for the vaginal and colostrum samples (Table 1) relative to all the metrics utilized.

Principal coordinate analysis (PCoA) was applied to compare the similarities among community membership (Figure 3). We observed a different pattern of microbial composition for each tissue sample. The colostrum samples had a similar microbial community structure to that of the vaginal samples; this was true in both the weighted (Figure 3A) and unweighted (Figure 3B) distance metrics. In contrast, the fecal samples shower fewer similarities to the microbial community of the colostrum samples; this indicated that the fecal and colostrum organisms formed separate groups. Some vaginal samples had a similar community composition to that of the fecal samples, while others showed greater similarity to the colostrum samples.

### 3.3. Correlation between Biochemical Biomarkers, Mammary Gland Health Status and Microbiomes

The results of a Spearman’s correlation analysis between the biochemical markers of negative energy balance, evidence of a healthy mammary gland and the makeup of the microbiomes of the feces, vagina and colostrum samples are shown in Figure 4. *p* values (*p* ≤ 0.05) indicate the significant associations that were detected in this study. The data are shown in Appendix A.

The correlational analysis showed an association only for specific genera within the Bacteria kingdom. In general, the *Thauera* and *Bacillus* genera showed positive correlations with the levels of NEFAs, the concentration of total protein, the body weight of the cows at calving and the score for the health of the mammary gland. However, these associations were only significant for the concentration of triglycerides. In contrast, these bacterial genera (*Thauera* and *Bacillus*) showed a negative correlation with milk production (milk quantity and duration of the dry period) and colostrum quality (including the percentage of macrophages/epithelial cells and colostrum volume). The genus *Pseudomonas* had a positive correlation with markers for energetic metabolism and the qualitative indicators of colostrum (its physical score in the cup). The correlation was significant for the level of cholesterol in plasma, the quantity of milk production in the previous lactation and the percentage of macrophages/epithelial cells in colostrum. On the other hand, *Pseudomonas* had a significantly negative correlation with the duration of the dry period.

At the family level, *Rhodothermaceae* was significantly positively correlated with the level of cholesterol in serum. *Peptostreptococcaceae* was negatively correlated with the duration of the previous lactation as well as with the percentage of macrophages/epithelial cells in colostrum (Appendix A). The association of fecal bacterial genera and the composite of health measures has a significantly negative correlation for members of the Eubacteria kingdom and the Archaea kingdom, including the genera *Bacteroides*, *Ruminococcus*, *Christensenellaceae* and *Methanobrevibacter*. This was observed most clearly for the indicators of energetic metabolism in this study. Finally, we found a positive correlation between the genus *Romboutsia* and the level of cholesterol, triglycerides, total protein and haptoglobin in serum (Appendix A). When the vaginal microbiome was assessed for correlations with the total set of variables studied here it was observed that, at the genus level, uncultivable rumen bacteria were significantly negatively correlated with the levels of BHB and NEFAs, as well as with the parity of cows. The genus *Roseburia* showed a similar set of relationships in the vaginal microbiome. In contrast, the genera *Escherichia*, *Romboutsia* and the *Archaea* genus, *Methanobrevibacter*, all showed a positive correlation with the level of cholesterol, triglycerides, total protein and albumin in serum. The genus *Pseudomonas* had a significant negative correlation with the indicators of energetic metabolites measured in this study, including the level of triglycerides, cholesterol, total protein, albumin and iron (Appendix A).

## 4. Discussion

It is our understanding that this is the first report which has attempted to examine the correlations between the indicators of mammary health with indicators of NEB as they relate to the microbiome of colostrum, the vagina and of feces from Holstein cows at the time of calving. These niches are all relevant to the health of cows and they all contribute to the transfer of microbes to calves.

Colostrum samples showed a great predominance of the phylum Proteobacteria. Other phyla of interest in colostrum were Firmicutes and Bacteroides, as were the genera *Pseudomonas*, *Staphylococcus*, *Acinetobacter* and *Stenotrophomonas*. This profile has also been described in other studies of the microbiome of bovine colostrum derived from healthy udders [34,35]. Although some of the bacterial genera found are considered to be potential pathogens of mastitis, clinical mastitis will only develop if mammary gland homeostasis is disturbed. None of these are considered to be overwhelming primary mammary pathogens [15]. Some authors have hypothesized that, at low numbers, these bacteria may be part of the normal milk microbiota; therefore, they may have other functions. They may function in aiding the regulation of the innate immune-mediated housekeeping functions of the mammary gland [15,16,17].

Other genera were also detected in colostrum: *Ochrobactrum* and *Thauera*. This is the first time that these genera were identified in bovine colostrum; however, these genera are associated with the environment of cows (water and soil) and are part of the denitrifying community of pasture [36]. Hence, finding these organisms in colostrum may be related to the housing (compost barn) used for the late pregnant cows during the last month of gestation on packed-earth flooring.

Similarly, the fecal microbiota observed in this study was similar to what has previously been reported in cattle. The bacterial genera most frequently observed were *Oscillospira*, *Christensenellaceae R-7* group, *Ruminococcus*, *Paeniclostridium* and *Bacteroides*. These findings are similar to those of the gastrointestinal microbiota of cattle reported by other authors [37,38]. In the vaginal microbiome the most abundant genera identified were *Pseudomonas*, *Christensenellaceae R-7*, *Paeniclostridium* and *Romboutsia*. These results were similar to previously reported studies on Holstein cows at the time of calving [13,18,19,35].

As to genus identification, *Acinetobacter* was the most frequently observed in the vaginal samples. This genus was previously observed in both fecal samples and in bulk tank milk samples [10,19,39]. The identification of this bacterial genus in vaginal samples may be a consequence of the anatomical proximity of the anus to the vulva [13,18,19,35].

Among the communities studied here, it was observed that the fecal samples had the greatest richness and diversity (alpha diversity) in microorganisms present of the three sites studied. The results were similar to those previously published for cows that were sampled near the time of calving. One study observed that the mean alpha diversity indices for maternal sources of microbiota indicated that the udder skin had the greatest richness and diversity of bacterial organisms [13]. This was followed by the vagina and less richness and diversity in the colostrum samples. Beta diversity documents the similarities or differences between microbial communities [40]. The UniFrac weighted and unweighted index were used to evaluate the distance between the colostrum, fecal and vaginal samples. Colostrum exhibited a smaller distance from some vaginal samples and a strongly different profile compared with fecal samples. These results can be reflective of the differences in alpha diversity that each type of sample presented. Likewise, the taxonomic profile showed that the fecal and vaginal samples had a greater abundance of Firmicutes, Proteobacteria and Bacteroidetes, while colostrum samples were dominated by the phylum Proteobacteria.

Some authors have offered the hypothesis that there is a possible endogenous colonization of the mammary gland by microorganisms from the gastrointestinal tract using an entero-mammary route in ruminants [17,41]. This hypothesis would explain the shared presence of a small number of OTU bacteria belonging to the genera *Ruminococcus* and *Bifidobacterium* as well as to the family *Peptostreptococcaceae* in milk samples [41]. In this investigation the colostrum samples showed a less similar microbial community to the fecal samples. In the analysis of correlations between the health of the cows and their microbiome it was observed that most of the associations were significantly negative between the parameters associated with energy metabolism relative to the taxa of *Pseudomonas*, *Christensenellaceae R-7* group and *Methanobrevibacter*, while *Escherichia* and *Romboutsia* had a positive correlation with these energy-associated markers.

Previous studies have described large changes in the relative abundances in the *Christensenellaceae* family (*Christensenellaceae group R-7* genus) around the time of calving in cows [8,42]. In ruminants *Christensenellaceae R-7* group play a key role in maintaining the GIT structure because of their role in generating the structural carbohydrate fermentation products, mainly acetic and butyric acid, required in ruminant metabolism. This is in addition to necessary syntrophic partnerships with *Methanobrevibacter* (the main methanogen in the GIT) [41,43]. An earlier study concluded that *Christensenellaceae* altered host gene expression and reduced inflammation during *E. coli* infection. Furthermore, this genus has been associated with the development of lean and healthy humans [44]. This observation could explain the negative relationship that these microorganisms have with the parameters we evaluated. On the other hand the positive association that the *Escherichia* presented with the same parameters would also make sense under this interpretation.

Archaea microorganisms of the *Methanobrevibacter* genus were found only in fecal and vaginal samples. These organisms showed both negative and positive associations with several biomarkers. They appeared to be associated with NEFAs, BHB, triglycerides and cholesterol. In ruminants the methanogens are extremely halophilic, thermophilic and anaerobic. This is due to their unique modes of energy metabolism and their primitive physiological regulation. Methanogens use a limited range of one-carbon compounds as substrates and convert them into methane [45].

*Ruminococcaceae* play a predominant role in ruminal biohydrogenation and is one of the most abundant families found in the samples from this research. Some authors have observed negative correlations between the genus *Ruminococcus* and milk production [46]. In this study no organisms of the genera belonging to this family were observed; thus, we were unable to generate any correlations. Our study showed a negative correlation between the *Ruminococcus* 2 genus and weight, cholesterol and triglyceride level among the fecal microbiome of cows at parturition. Another study showed an increase in the relative abundance of the genus *Ruminococcus* 2 in cows after calving in the ruminal microbiota [47]. Similarly, another study reported negative correlations between feed efficiency relative to dry matter intake and the relative abundances of *Ruminococcus* and *Methanobrevibacter* in the rumen of peripartum cows [48]. These results support the notion that calving is a critical time in the productive life of cattle. The nutritional and metabolic modifications that occur during the periparturient period in cows appear to modify the composition and function of the microbiota. Alterations in the GIT microbiome can in turn affect the balance of systemic metabolism. However, more research is needed to understand the role of the microorganisms identified here in the health, physiological function and regulation of metabolism of cows at the time of calving.

## 5. Conclusions

The data collected by analyzing the microbiome in samples of colostrum, feces and the vagina from twenty cows at calving allowed us to conclude that the fecal and vaginal microbiota have more shared elements in their composition than with the microbiota of colostrum. In this study colostrum had a unique microbial profile relative to the other sites. It is inferred that the vaginal microbiota was strongly influenced by a contribution from that of feces due to the anatomical proximity of the sites and the messy nature of the physical processes of calving. In contrast, colostrum appears to have its own microbial profile. It appears to have a strong contribution from the external environment, including the soil in the bedding of the cows. Most of the correlations observed during this investigation were negative. When the microbiota was associated with several markers of energetic metabolism and assessed against the health of the mammary gland, most of the correlations were negative. Associations were detected for some specific microbial elements relative to NEFAs, cholesterol, triglycerides and prior lactation production of milk by the cows’ indicators of energetic metabolism. Therefore, we have concluded that there appear to be relationships between the microorganisms that colonize different mucosal tissue structures of cows, composed primarily of commensals that are linked to systemic energy metabolism, and the health of the mammary gland of cows at the time of calving.

## Figures and Tables

**Figure 1 animals-12-00003-f001:**
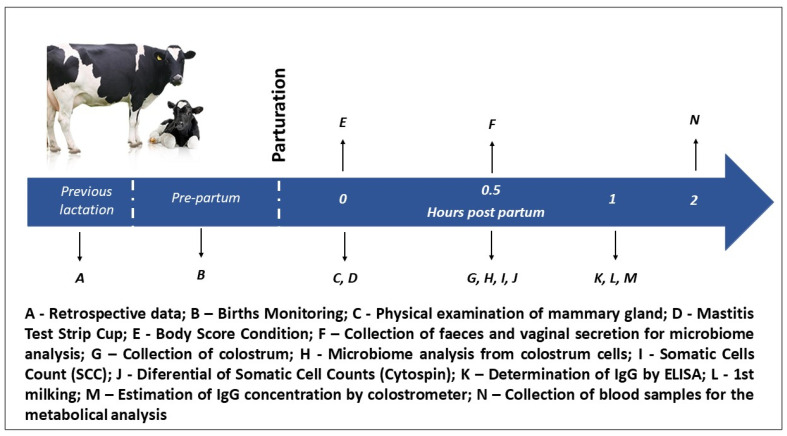
Timeline indicating the order and timing of observations, sample collections and interventions carried out during the course of the study.

**Figure 2 animals-12-00003-f002:**
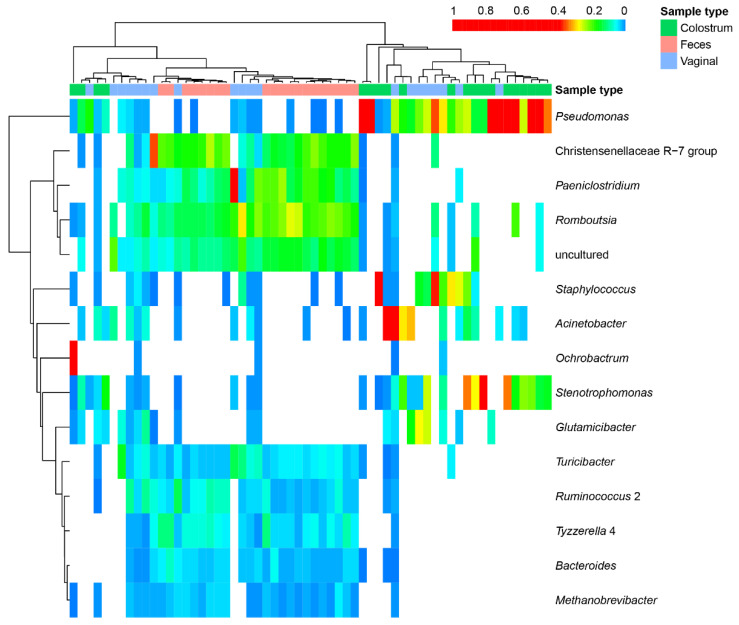
A heat map showing the major bacterial taxa found in samples collected from 20 Holstein cows at parturition. Each row refers to a taxon (genus) and each column represents a sample (as listed in code below the map). The color legend at the top of the graph indicates the type of sample evaluated (colostrum, fecal or vaginal).

**Figure 3 animals-12-00003-f003:**
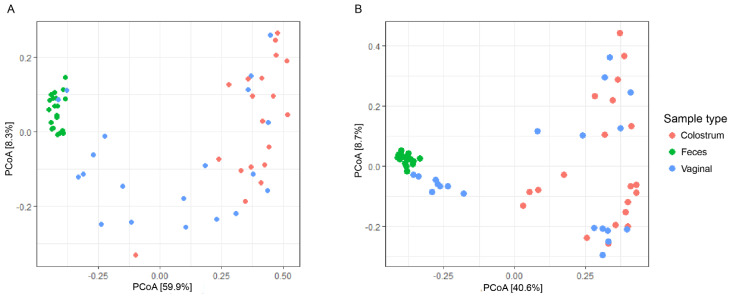
A summary of the global microbiome analysis of the samples from the 20 Holstein cows taken immediately after calving are displayed using weighted (**A**) and unweighted (**B**) UniFrac metric calculations to plot distances. The mucosal source of the samples is indicated by the color of the marker.

**Figure 4 animals-12-00003-f004:**
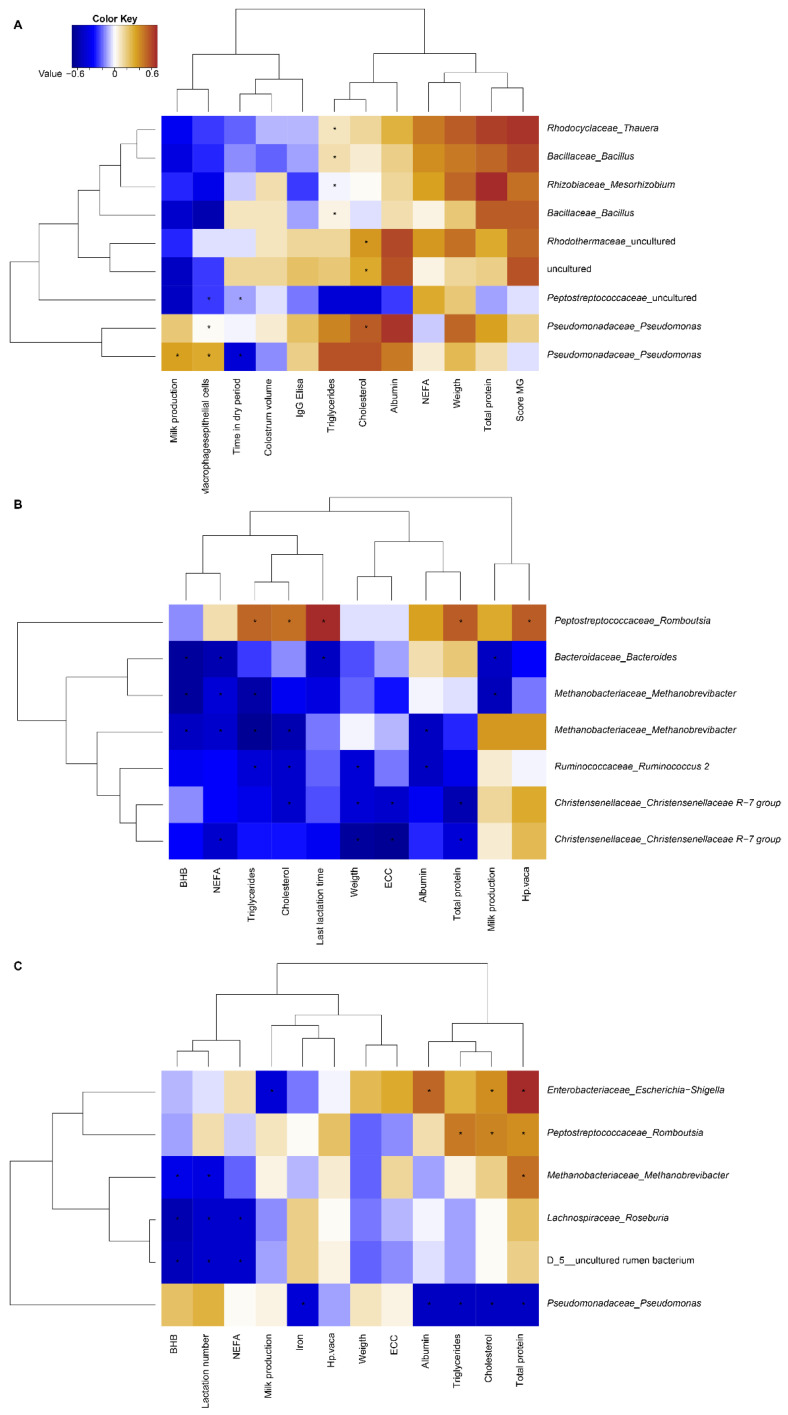
A display of the Spearman correlation between microbiome bacteria and the group of health variables collected. The correlations between (**A**) colostrum bacteria; (**B**) vaginal bacteria; and (**C**) vaginal bacteria against the health variables are indicated by colors (brown: positive; blue: negative). Significant correlations (*p* ≤ 0.05) are indicated by *; only the specific health variables that had at least one significant correlation with bacteria are shown in these maps.

**Table 1 animals-12-00003-t001:** Species richness is represented based on an analysis using the Chao1 score, the generation of Shannon diversity values and the qualitative observed index for samples collected from the 20 Holstein cows immediately after calving. The data are presented as the means and standard deviations of the values from all 20 cows by tissue.

Samples Type	Chao1	Shannon	Observed
Colostrum	53.3 (8.1)	1.7 (1.06)	42.2 (25.9)
Feces	173.7 (14.5)	3.88 (0.60)	137.1 (30.58)
Vaginal secretion	136.3 (12.1)	3.67 (0.19)	109.1 (9.4)

## Data Availability

The nucleotide sequence data reported are available in the NCBI under BioProject accession number PRJNA790039.

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
