# Peer review of "Relationships among Indicators of Metabolism, Mammary Health and the Microbiomes of Periparturient Holstein Cows"

_animals, 2021, doi:10.3390/ani12010003_

Round 1

Reviewer 1 Report

The authors conducted a study to investigate potential relationships between health parameters during the periparturient period and the microbiome of the digestive tract (via fecal samples), vagina, and colostrum in Holstein cows. While I believe there is some merit to this research, I ultimately concluded in a 'reject' recommendation because the results were poorly presented and there were numerous grammatical issues. Highlighted below are just some of the minor and major issues: 

Minor issues:

  • In the title, 'Holstein' should be capitalized. 
  • Remove all plural first-person pronouns (e.g., we, us, our).
  • Line 85: Use the term "parturition" not "delivery". 
  • Line 101: Reference for estimating body weight by thoracic perimeter. Also, remove additional closed parenthesis.  
  • Line 297: NEFA not MEFA

Major issues:

  • The grammar was very poor, particularly in the 'Simple Summary' and 'Abstract' sections. Extensive edits are needed throughout the paper though. 
  • I personally do not like the use of 'maternal mucosal'. This could be appropriate when describing the vaginal microbiome, but would be a greater stretch for the colostrum and fecal microbiomes.  
  • Line 81-82: You state 'several published works', but only provide one reference? 
  • Line 82-83: I disagree with this statement. Please rephrase for the scope of this study. 
  • Providing a timeline or at least indicating specifically when samples were taken would be helpful. 
  • It might be helpful to have retrospective treatment groups when conducting and presenting the results. For example, classifying cows based on their NEB status using the physiological parameters and then evaluate differences in the microbiomes.  
  • For Figure 1, is this relative abundance of the bacteria? Many microbiome studies will also present the relative abundance data and focus on the bacteria that are greater than 1% relative abundance. This helps focus on specific bacteria that many actually be playing a role. 
  • For Table 1, are these alpha-diversity numbers correct?? Typically, Choa1 and and Observed are much greater (in the hundreds and/or thousands) because Chao1 is the estimate of the total number of species present and Observed is the number of observed species. 

Author Response

First of all, we would like to thank the editors/reviewers of this study who presented valuable suggestions and very pertinent comments, thus contributing to improve the quality of our manuscript. After a detailed analysis of the comments and questions, as well as the errors pointed out and suggestions contained in the opinions sent to us, the article has undergone some changes, which are indicated below.

General comments

The authors conducted a study to investigate potential relationships between health parameters during the periparturient period and the microbiome of the digestive tract (via fecal samples), vagina, and colostrum in Holstein cows.  While I believe there is some merit to this research, I ultimately concluded in a 'reject' recommendation because the results were poorly presented and there were numerous grammatical issues. Highlighted below are just some of the minor and major issues.

Minor issues:

In the title, 'Holstein' should be capitalized.

R: Thank you for your observation. It has been checked and corrected.

Remove all plural first-person pronouns (e.g., we, us, our). Use the term "parturition" not "delivery".

R: It has been checked and corrected.

Line 101: Reference for estimating body weight by thoracic perimeter. Also, remove additional closed parenthesis.

R: The method used was that described by Heinrichs and Hargrove, 1987. Reference was added to the manuscript as follows:

  1. Heinrichs, A.J.; Hargrove, G.L. Standards of weight and height for Holstein heifers. J Dairy Sci. 1987, 70, 653–660.

 The additional closed parenthesis was checked and removed.

Line 297: NEFA not MEFA

R: It has been checked and corrected.

Major issues:

The grammar was very poor, particularly in the 'Simple Summary' and 'Abstract' sections. Extensive edits are needed throughout the paper though. I personally do not like the use of 'maternal mucosal'. This could be appropriate when describing the vaginal microbiome, but would be a greater stretch for the colostrum and fecal microbiomes

R: All of grammatical errors were corrected in the body of our manuscript revised version. In addition, the 'maternal mucosal' concept was corrected.

Line 81-82: You state 'several published works', but only provide one reference?

R: It has been checked and corrected. Missing references in the manuscript were added.

Line 82-83: I disagree with this statement. Please rephrase for the scope of this study.

R: It has been checked and corrected. “However, there are few studies that describe the maternal microbiota at the time of calving and its relationship with health parameters associated with this process”

Providing a timeline or at least indicating specifically when samples were taken would be helpful.

R: In order to improve the understanding of the collection of samples and data, a figure (Figure 1) that describes the methodology used in this work was added to the manuscript.

It might be helpful to have retrospective treatment groups when conducting and presenting the results. For example, classifying cows based on their NEB status using the physiological parameters and then evaluate differences in the microbiomes

R: When analyzing the data of this study, an attempt was made to create status according to the different concentrations measured in the blood, the health status of the mammary gland and retrospective data collected. However, due to the low number of samples (n = 20) and the fact of selecting apparently healthy cows, the results of these analyzes did not show statistically significant variations.

For Figure 1, is this relative abundance of the bacteria? Many microbiome studies will also present the relative abundance data and focus on the bacteria that are greater than 1% relative abundance. This helps focus on specific bacteria that many actually be playing a role

R: Figure 1 (figure 2 in the corrected version) shows the proportion of the 15 most abundant OTUs in colostrum, fecal and vaginal samples. This, in order to see the clustering of these samples. The heatmap was built using Ward’s hierarchical clustering method (ward.d2) and the OTUs were filtered with minimum threshold = 1000.

In addition, the results of this figure allow us to visualize that there is a similarity in OTU and abundance between the fecal microbiomes and a group of vaginal samples. On the other hand, it is shown that another group of vaginal samples is similar to the microbial profile of colostrum.

For Table 1, are these alpha-diversity numbers, correct?? Typically, Chao1 and and Observed are much greater (in the hundreds and/or thousands) because Chao1 is the estimate of the total number of species present and Observed is the number of observed species.

R: It has been checked and corrected. Indeed, there was an error at the time of rarefaction, the results were corrected in the manuscript with the correct rarefaction (min 1000)

Reviewer 2 Report

This was a very interesting paper.

Seems like you have 2 objectives the first is line83-85 and the 2nd is line 86-87.

Line 87 to 89 is not need in the introduction it belongs in material and methods section

Line 101 was the thoracic measurement done with a tape if so state  which one.

Line 172 and 173 remove "of blood".

Lines 295-305 round the numbers so they are what you measured, ie for BW if you can only measure 607 do not report 607.00

Figure 1 is hard to read, can it be made into a table, if not can it be made larger.

Figure 3 is also hard to read can it be made larger

Line 434-435 this is awkward can you please rewrite

Author Response

First of all, we would like to thank the editors/reviewers of this study who presented valuable suggestions and very pertinent comments, thus contributing to improve the quality of our manuscript. After a detailed analysis of the comments and questions, as well as the errors pointed out and suggestions contained in the opinions sent to us, the article has undergone some changes, which are indicated below.

General comments

This was a very interesting paper.

R: Thank you for your comments.

Specific comments:

Seems like you have 2 objectives the first is line83-85 and the 2nd is line 86-87.

R: It has been reviewed, the paragraph changed and the manuscript corrected.

Line 87 to 89 is not need in the introduction it belongs in material and methods section

R: The method used was that described by Heinrichs and Hargrove, 1987. Reference was added to the manuscript as follows:

Heinrichs, A.J.; Hargrove, G.L. Standards of weight and height for Holstein heifers. J Dairy Sci. 1987, 70, 653–660.

Line 172 and 173 remove "of blood".

R: It has been checked and corrected.

Lines 295-305 round the numbers so they are what you measured, ie for BW if you can only measure 607 do not report 607.00

R: It has been checked and corrected in manuscript

Figure 1 is hard to read, can it be made into a table, if not can it be made larger. Figure 3 is also hard to read can it be made larger

R: Figure 1 and 3 (figure 2 and 3 in the corrected version) was modified to improve its visualization

Line 434-435 this is awkward can you please rewrite

R: It has been checked and corrected in manuscript. “The hypothesis of some authors made that there is a possible endogenous colonization of the mammary gland by microorganisms from the gastrointestinal tract has been proposed as an entero-mammary route in ruminants [28, 36]. This hypothesis would ex-plain the shared presence of a small number of OTU bacteria belonging to the genera Ruminococcus and Bifidobacterium and to the family Peptostreptococcaceae in milk samples” [36].

Reviewer 3 Report

General

Comments

While this paper is the result of considerable work and makes a novel contribution to the literature on the relationship between the microbial population of various physiological materials and dairy cow health, there are several questions that should be addressed.  First, the rationale for sampling feces, vaginal fluid, and colostrum is unclear.  Since the paper is dealing with energy metabolism, wouldn’t of samples from the rumen and intestine have been more useful.  Furthermore, while the paper repeatedly mentions the analysis of the mucosa, sampling of feces and colostrum would seem to be a rather crude approach to sampling the mucosa of those organs.  Second, while the emphasis of the paper is on the negative energy balance of the cows, measurements related to energy balance were weak or unclear.  A single body weight or previous milk production doesn’t have much value particularly when feed intake does not seem to be measured.  The measurement of metabolites in the blood is more useful, but it is unclear when those samples were taken and taking more than a single measurement would have been useful.

Line

3

‘Holstein’ should be capitalized

21

As ‘appear’ refers for a visual image, ‘seem’ is a more correct word.   Regardless, the concept is not very definitive

25

The statement that ‘the microbiome on different maternal mucosa are different’ seems obvious.

25-27

What is the rationale for this conclusion?

29

Change ‘this’ to ‘thus’

31

What was the basis of selecting the mammary gland, vagina and feces selected and not the rumen or intestine?

35

The statement that ‘the microbiome on different maternal mucosa were different’  seems obvious. A more specific statement would be desirable.

36

What are the ‘products that define NEB’ that were measured?

39

Change ‘re-lationships’ to ‘relationships’

83-85

While it is assumed that the measurements of the colostrum are related to ‘some of the health parameters’ mentioned by the authors, it is not clear that it is.  Furthermore, the basis of the colostrum measurements is not described in the introduction.

104-113

Were cow body weight, feed intake, and milk production measured to establish energy balance?  If not, why?

126-143

While the procedures for collection and analysis of colostrum seem correct, the rationale for its collection and analysis is unclear.

145-146

The rationale for collecting samples from the vagina is unclear.  Similarly, because this experiment is supposed to deal with mucosa, the collection of feces is questionable.

152

Change ‘The’ to ‘Each’

156-163

The rationale for collection of colostrum is unclear.

167

It is unclear when and how frequently blood samples were collected in relation to the time of calving.  The way it is written, blood seems to have been sampled at one undefined time which would be inadequate.

192-236

Again, the rationale for the analysis of colostrum is unclear.

295

Since changes in body condition, feed intake, and milk production were not measured, it is difficult to establish the degree of negative energy balance. This is supported by the metabolic measurements that seem to should a wide variation in the degree of NEB across cows.

297

Change ‘MEFA’ to ‘NEFA’

349

Since change in body weight was not measured, body weight by itself is not of much value.

467

Body weight without weight change is of little value.

472-473

How do the results of this experiment support the statement that ‘These results support that calving is a critical time in the productive life of cattle’?

490-496

These relationships seem rather weak to conclude that they were linked to systemic energy metabolism and mammary health.

494

Throughout the paper, the authors imply that they were dealing with the microbial population of the mucosa of different organs.  However, sampling feces and colostrum would seem to be a crude approach from assaying the microbial population of the mucosa of the rectum and mammary gland.

Author Response

First of all, we would like to thank the editors/reviewers of this study who presented valuable suggestions and very pertinent comments, thus contributing to improve the quality of our manuscript. After a detailed analysis of the comments and questions, as well as the errors pointed out and suggestions contained in the opinions sent to us, the article has undergone some changes, which are indicated below.

General comments

While this paper is the result of considerable work and makes a novel contribution to the literature on the relationship between the microbial population of various physiological materials and dairy cow health, there are several questions that should be addressed. First, the rationale for sampling feces, vaginal fluid, and colostrum is unclear. Since the paper is dealing with energy metabolism, wouldn’t of samples from the rumen and intestine have been more useful. Furthermore, while the paper repeatedly mentions the analysis of the mucosa, sampling of feces and colostrum would seem to be a rather crude approach to sampling the mucosa of those organs. Second, while the emphasis of the paper is on the negative energy balance of the cows, measurements related to energy balance were weak or unclear. A single body weight or previous milk production doesn’t have much value particularly when feed intake does not seem to be measured. The measurement of metabolites in the blood is more useful, but itis unclear when those samples were taken and taking more than a single measurement would have been useful.

R: Thank you for your comments, they will certainly help to improve the quality of the paper. Below the corrections in details

Specific comments:

Line 3 Holstein’ should be capitalized

R: It has been corrected.

Line 21. As ‘appear’ refers for a visual image, ‘seem’ isa more correct word. Regardless, the concept is not very definitive

R: It has been checked and corrected.

Line 25. The statement that ‘the microbiome on different maternal mucosa is different’ seems obvious.

R: It has been checked and corrected in introduction

Line 25-27. What is the rationale for this conclusion?

R: The conclusion was changed for “From these results, the conclusion that the microorganisms colonizing different mucosal tissues of the cow are linked to systemic energy metabolism of the cow, and have an impact on the health of the mammary gland of the host following calving were drawn”

We agree with all of your comments that measuring NEFA and BHB only at calving do not represent a negative energetic balance or ketosis, so we changed our title, conclusion and all content of the whole text

Line 29. Change ‘this’ to ‘thus’.

R: It has been checked and corrected.

Line 31. What was the basis of selecting the mammary gland, vagina and feces selected and not the rumen or intestine?

R: This work is part of a larger project that evaluates the influence of the maternal microbiota on the colonization of the intestinal microbiome and immune development of calves in the first month of life. Furthermore, the main objective of this paper was to describe the correlation between some health parameters and the cow's microbiome at the time of calving. In this context, the mammary gland (colostrum), fecal and vaginal samples were selected as adequate and significant to represent the parturition event.

Line 35. The statement that ‘the microbiome on different maternal mucosa were different’ seems obvious. A more specific statement would be desirable.

Line 36. What are the ‘products that define NEB’ that were measured?

R: In this study mainly, significant negative associations between the products that define NEB (NEFA, BHB, triglycerides and cholesterol). This was added to the manuscript (Line 36)

Line 39. Change ‘re-lationships’ to ‘relationships’

R: It has been checked and corrected.

Line 83-85. While it is assumed that the measurements of the colostrum are related to ‘some of the health parameters’ mentioned by the authors, it is not clear that it is. Furthermore, the basis of the colostrum measurements is not described in the introduction.

R: It has been checked and added in Introduction. “Because of the metabolic requirements of the gland, particularly in early lactation, it has been suggested that the cow should be considered an appendage to the mammary gland, rather than vice versa [5]. In the last third of pregnancy there are alterations of the secretory capacity of the mammary epithelial cells, the decrease in parenchyma occurs gradually during lactation, this decrease may be reduced to the metabolic conflict between pregnancy and lactation. A nonlactating period is necessary for optimal milk production in the succeeding lactation is important for the replacement of senescent and damaged cells of the mammary epithelium. [6]. Replacement cells may be responsible for the expansion and maintenance of the secretory cells of the mammary gland that influence colostrogenesis and milk production in subsequent lactation [6,7].”

Line 104-113. Were cow body weight, feed intake, and milk production measured to establish energy balance? If not, why?

R: The parameters mentioned were not measured, since the study aimed to describe what was happening only at the time of calving in apparently healthy cows.

Line 126-143. While the procedures for collection and analysis of colostrum seem correct, the rationale for its collection and analysis is unclear.

  1. This fact was justified and corrected in Lines 31 and line 83-85

Line 152. Change ‘The’ to ‘Each’

R: It has been checked and corrected.

153-166. The rationale for collection of colostrum is unclear.

R: It has been checked and added in Introduction.

Line 167. It is unclear when and how frequently blood samples were collected in relation to the time of calving. The way it is written, blood seems to have been sampled at one undefined time which would be inadequate.

R: Blood samples were collected up to 2 hours after calving.

A considerable limitation of this research was the moment at which these metabolic biomarkers were measured, mainly in the BHB concentration. Samples for BHB measurement should be collected between 3 to 50 days postpartum. The plasma concentration of NEFA begins to increase two to four days before calving, but in animals at risk of metabolic disorders it may begin to increase earlier. In relation to the rest of the parameters evaluated, all showed variation of the cohort points established in the literature, however these alterations are the result of a physiological process, which is parturition.

As the objective of this study was to relate the microbiome with the indicator parameters of NEB at the time of calving, these data were not discussed in the manuscript.

Line 192-236. Again, the rationale for the analysis of colostrum is unclear.

R: This work is part of a larger project that evaluates the influence of the maternal microbiota on the colonization of the intestinal microbiome and immune development of calves in the first month of life. Furthermore, the main objective of this paper was to describe the correlation between some health parameters and the cow's microbiome at the time of calving. In this context, the mammary gland (colostrum), fecal and vaginal samples were selected as adequate and significant to represent the parturition event.

Line 295. Since changes in body condition, feed intake, and milk production were not measured, it is difficult to establish the degree of negative energy balance. This is supported by the metabolic measurements that seem to should a wide variation in the degree of NEB across cows.

R: We agree with this statement, the term NEB was replaced by "energy metabolism markers" throughout the manuscript (including title), since using parameters such as NEFA and BHB at the time of calving is not decisive to measure NEB.

Line 297. Change ‘MEFA’ to ‘NEFA’

R: It has been checked and corrected.

Line 349. Since change in body weight was not measured, body weight by itself is not of much value. Line 467. Body weight without weight change is of little value.

R: We agree with you. The context of this parameter was changed in the text, since our focus was on showing a "photograph" of the calving moment.

Line 472-473. How do the results of this experiment support the statement that ‘These results support that calving is a critical time in the productive life of cattle’?

R: The paragraph refers to the article cited above:

"The nutritional and metabolic modifications that occur during the peri-parturient period in the cow appear to modify the composition and function of the microbiota. Alterations in the GIT microbiome can in turn affect the balance of systemic metabolism".

            Delgado B.; Bach A.; Guasch I.; González C.; Elcoso G.; Pryce J.E.; González-Recio O. Whole rumen metagenome sequencing allows classifying and predicting feed efficiency and intake levels in cattle. Sci. Rep. 2019, 11.

Line 490-496. These relationships seem rather weak to conclude that they were linked to systemic energy metabolism and mammary health.

R: The relationship between energy balance and immune response in the general concept and in the mammary gland have been postulated in literature. However, the main reason that we evaluated the microbiome from gut (represented by fecal microbiota) and mammary gland was the possible communication between these mucosal system by the entero-mammary way postulated by the similarity between the human microbiome from fecal samples from dams, colostrum and fecal samples from offspring. In our research this way can be considered null, since the microbiome from these two regions had different profiles.

Line 494. Throughout the paper, the authors imply that they were dealing with the microbial population of the mucosa of different organs. However, sampling feces and colostrum would seem to be a crude approach from assaying the microbial population of the mucosa of the rectum and mammary gland.

R: This is true, the concept "mucosa" for colostrum and fecal samples is a crude term. To resolve this question, it was decided to change these concepts throughout the manuscript by: "colostrum, fecal and vaginal samples"

Reviewer 4 Report

This paper aims to investigate the correlation between the microbiome composition, systemic host metabolic balance, and some health parameters of cows immediately after calving.

The subject is interesting and adds information on a very complex topic. The paper is quite innovative and interesting although in my opinion the authors need to improve the introduction by adding more information relative to the complexity of transition period in dairy cow. Moreover, they should improve and detail material and methods.

My recommendation is Moderate Revision

Specific comments:

L3: In the title replace mammary with udder

L22:  which makes important understand these events

L25: …mucosal areas…

L35: …mucosal areas…

L39: relationships

L45-57: Overall, the characterization and the impact of NEB and the complex physiological and endocrine changes during transition period have to be better described and better supported by updated references, including also health and reproductive consequences.

L87-89: Remove this sentence because is part of material and methods and not of the aims of the project

L101: do you mean heart girth or wither girth?

L109-111: It's not clear what you want to say. Please, rephrase. Did you use a TMR?

L112-113: did it met or exceed the requirements based on what? body weight? Please, specify

L113: Put NRC in the references.

L116: “:”

L117: add references

L118: bedding

L119: Were calving pens shred by different cows in the same time or were they single pens? Please specify and better describe the pens, (e.g. size).

L121: heart girth or wither girth

L125: you need to specify when did you sample colostrum, how many hours after calving. The same for the collection of the other samples and the measure of the various parameters.

L126: each abbreviation must be explained at first use

L128-129: It's not clear. After cleaning the udder did you use swabs to measure microbiological quality of the udder skin before collecting colostrum in order to guarantee its microbiological quality?? Please rephrase and specify.

Author Response

First of all, we would like to thank the editors/reviewers of this study who presented valuable suggestions and very pertinent comments, thus contributing to improve the quality of our manuscript. After a detailed analysis of the comments and questions, as well as the errors pointed out and suggestions contained in the opinions sent to us, the article has undergone some changes, which are indicated below.

This paper aims to investigate the correlation between the microbiome composition, systemic host metabolic balance, and some health parameters of cows immediately after calving.

The subject is interesting and adds information on a very complex topic. The paper is quite innovative and interesting although in my opinion the authors need to improve the introduction by adding more information relative to the complexity of transition period in dairy cow. Moreover, they should improve and detail material and methods.

My recommendation is Moderate Revision

R: Thank you for your comments, the corrections will be detailed below in the specific comments.

Specific comments:

L3: In the title replace mammary with udder. L22: which makes important understand these events. L25: ...mucosal areas...L35: ...mucosal areas...L39: relationships

R: It has been checked and corrected.

L45-57: Overall, the characterization and the impact of NEB and the complex physiological and endocrine changes during transition period have to be better described and better supported by updated references, including also health and reproductive consequences.

R: It has been checked and corrected in manuscript.Immunosuppression predisposes cows to a retained placenta and infectious diseases, such as mastitis and metritis. Transition cow health disorders are associated with negative outcomes in terms of milk production loss, reduced reproductive performance, risk of involuntary culling and loss of profitability due to the direct and indirect costs associated with the disorder [3,4]. While the negative outcomes related to milk yield, reproduction, and profit have been extensively documented, the relationships between transition cow health disorders and feed efficiency have not been investigated thoroughly [4]”

L87-89: Remove this sentence because is part of material and methods and not of the aims of the project.

R: It has been checked and corrected.

L101: do you mean heart girth or wither girth?

R: We used heart girth. It has been corrected in manuscript

L109-111: It's not clear what you want to say. Please, rephrase. Did you use a TMR?

R: The composition of the diet was managed in total mixed ration (TMR) and mainly with forage and corn silage as base material. Minerals and vitamins were added during the feed manufacturing process. Supplemental water and mineral salts were provided ad libitum

L112-113: did it met or exceed the requirements based on what? body weight? Please, specify

L113: Put NRC in the references. L116: “:” L117: add references. L118: bedding.

R: It has been checked and corrected.

L119: Were calving pens shred by different cows in the same time or were they single pens? Please specify and better describe the pens, (e.g. size).

R: The calving pen had a capacity for 2 cows, however in this work, there were no simultaneous calvings. Therefore, the cleaning of the bed material change was carried out after each delivery (individual).

L121: heart girth or wither girth

R: It has been corrected.

L125: you need to specify when did you sample colostrum, how many hours after calving. The same for the collection of the other samples and the measure of the various parameters.

R: L122... At birth, the weight of the cow was estimated using the thoracic diameter (heart girth) [14]. L127… Immediately after calving (maximum 1 hour postpartum), an initial cleaning and disinfection. L146, L147… Vaginal and fecal samples were collected following the cleaning and disinfection of the external recto-vulvar region, after colostrum samples collected.

In order to improve the understanding of the collection of samples and data, a figure (Figure 1) that describes the methodology used in this work was added to the manuscript.

L126: each abbreviation must be explained at first use

R: It has been corrected. “Standard operating procedures (SOP)”.

L128-129: It's not clear. After cleaning the udder did you use swabs to measure microbiological quality of the udder skin before collecting colostrum in order to guarantee its microbiological quality?? Please rephrase and specify

R: It has been corrected. “The udder was initially cleaned of mud and feces using a soft brush. This was immediately followed by teat antisepsis. This were dipped in a 2% chlorine bleach solution and dried with individual clean paper towels. The teat ends were sprayed with povidone iodine solution (Riodeine®, Rioquimica) then dried with sterile gauze. Finally, antiseptic treatment of the teat ends involved rubbing sterile gauze saturated with ethanol 70% over the surface.
The teats were stripped of the first jets of mammary secretion. During this procedure, a physical examination of the mammary gland was conducted by visual inspection and a palpation of gland structures [13]”.

Round 2

Reviewer 1 Report

The authors did address some issues, but in doing so, they created many more grammatical issues. There are numerous run-on sentences, misspelled words (example, Line 49 'serious'), words with dashes that don't require them, etc. Additionally, reference numbers are not used correctly (example, Line 92). Much of the wording and grammar need to be addressed before an extensive review can be done. 

Author Response

We thank the reviewer for his/her specific comments. Our paper had a great improvement after all changes after reviewrs suggestions. Paper have been submitted to a extensive eddition of English language and style by Dr David John Hurley, american professor and co-author of our paper. I hope that this new version to attend the english quality required by teh Animals Journal. Sincerelly. Prof. Viviani Gomes